# Spatial and temporal clustering of anti-SARS-CoV-2 antibodies in Illinois household cats, 2021–2023

Chi Chen[1], Mathias Martins[2], Mohammed Nooruzzaman[2], Dipankar Yettapu[1], Diego G. Diel[2], Jennifer M. Reinhart[3], Ashlee Urbasic[4], Hannah Robinson[4], Csaba Varga[1,5]*, Ying Fang[1,5]*

1 Department of Pathobiology, College of Veterinary Medicine, University of Illinois at Urbana-Champaign, Urbana, Illinois, United States of America, 2 Department of Population Medicine and Diagnostic Sciences, College of Veterinary Medicine, Cornell University, Ithaca, New York, United States of America, 3 Department of Veterinary Clinical Medicine, College of Veterinary Medicine, University of Illinois at Urbana-Champaign, Urbana, Illinois, United States of America, 4 Veterinary Diagnostic Laboratory at Veterinary Specialty Center, University of Illinois at Urbana-Champaign, Buffalo Grove, Illinois, United States of America, 5 Carl R. Woese Institute for Genomic Biology, University of Illinois at Urbana-Champaign, Urbana, Illinois, United States of America

* cvarga@illinois.edu (CV); yingf@illinois.edu (YF)

**Data Availability Statement:** All relevant data are within the manuscript and its Supporting Information files.

**Funding:** This project was supported by the National Institute of Health [Grant #R01AI166791,

## Abstract

This study aimed to evaluate the seroprevalence and spatial and temporal clustering of SARS-CoV-2 antibodies in household cats within 63 counties in Illinois from October 2021 to May 2023. The analysis followed a stepwise approach. First, in a choropleth point map, we illustrated the distribution of county-level seroprevalence of SARS-CoV-2 antibodies. Next, spatial interpolation was used to predict the seroprevalence in counties without recorded data. Global and local clustering methods were used to identify the extent of clustering and the counties with high or low seroprevalence, respectively. Next, temporal, spatial, and space-time scan statistic was used to identify periods and counties with higher-than-expected seroprevalence. In the last step, to identify more distinct areas in counties with high seroprevalence, city-level analysis was conducted to identify temporal and space-time clusters. Among 1,715 samples tested by serological assays, 244 samples (14%) tested positive. Young cats had higher seropositivity than older cats, and the third quarter of the year had the highest odds of seropositivity. Three county-level space-time clusters with higher-than-expected seroprevalence were identified in the northeastern, central-east, and southwest regions of Illinois, occurring between June and October 2022. In the city-level analysis, 2 space-time clusters were identified in Chicago's downtown and the southwestern suburbs of Chicago between June and September 2022. Our results suggest that the high density of humans and cats in large cities such as Chicago, might play a role in the transmission and clustering of SARS-CoV-2. Our study provides an in-depth analysis of SARS-CoV-2 epidemiology in Illinois household cats, which will aid in COVID-19 control and prevention.

to Y. Fang (PI) and D. Diel (MPI)]. The funders had no role in study design, data collection, and analysis, decision to publish, or preparation of the manuscript.

**Competing interests:** The authors have declared that no competing interests exist.

## Introduction

Severe acute respiratory syndrome coronavirus 2 (SARS-CoV-2), the causative agent of Coronavirus Disease 2019 (COVID-19), is a novel virus that emerged in late 2019 in Wuhan, China [1–3]. It belongs to the family *Coronaviridae* within the order *Nidovirales*, a group of positive-stranded RNA viruses with unique characteristics [4, 5]. SARS-CoV-2 has a broad host range and is the seventh coronavirus that could infect humans [6–8]. SARS-CoV-2 entry into host cells is initiated by binding to the angiotensin-converting enzyme 2 (ACE2) receptor through its spike (S) protein. ACE2 receptors in some animals are similar to those in humans, leading to increased susceptibility to the virus [9]. A wide range of animal species have been reported to be infected by SARS-CoV-2, including cats, dogs, deer, mink, and hamsters [10–14]. This has raised concerns about potential reservoirs in animal populations, which could lead to further spillover events and threats to public health.

SARS-CoV-2 has been reported to be transmitted from humans to animals, including domestic cats and dogs [15–18]. Previous studies showed that SARS-CoV-2 did not replicate well in dogs, but was able to efficiently replicate in domestic cats and further transmit the virus to naive contact animals [19]. A recent report provided evidence that domestic cats could transmit SARS-CoV-2 to humans [15]. Since domestic household cats have close contact with humans, it is important to understand the prevalence of SARS-CoV-2 exposure in household cat populations. Recent serological surveillance in cats reported a seropositivity range of 15% in China, 1% in Germany, and 5% in Portugal [20–22]. However, the prevalence of SARS-CoV-2 in domestic household cats has not been well-defined in the United States of America.

The state of Illinois is located in the midwest of the US and is ranked 6[th] by population size among the 50 US states, having an estimated 12,582,032 residents in 2022 [23]. The largest city in Illinois, Chicago, is ranked 3[rd] in the US by population size and had an estimated 2,665,039 residents in 2022 [24]. The city of Chicago is a domestic and international transportation center. The Port of Chicago is a busy port facility that has access to the Atlantic Ocean through the Great Lakes and Saint Lawrence Seaway. Moreover, Chicago's O'Hare International Airport, the 4[th] busiest airport in the world [25], serves as a central link for international and domestic travel. The unique geographic location and global transportation facilities make Illinois and Chicago potential hotspots for the transmission of pathogens.

In this study, we investigated the seroprevalence of SARS-CoV-2 antibodies in domestic cats in Illinois from October 2021 to May 2023. Spatial, temporal, and space-time scan statistical models were used to identify county-level clusters of higher-than-expected antibody-positive cats in Illinois. In counties identified with high seroprevalence, we conducted a city-level analysis to identify more refined space-time clusters. A logistic regression model was constructed to evaluate the impact of age and season on the odds of SARS-CoV-2 seroprevalence in domestic cats. The prevalence, temporality, and space-time clustering of SARS-CoV-2 infection in household cats in Illinois, and the city of Chicago, could guide future COVID-19 prevention and control programs.

## Materials and methods

### Data sources

The study period extended from February 2021 to May 2023, which included 1715 serum samples from household cats in 63 counties across Illinois. Samples were obtained from the Clinical Pathology Laboratory at the University of Illinois Veterinary Diagnostic Laboratory (Urbana, IL) and its satellite lab at the Veterinary Specialty Center (Bannockburn, IL). Samples consisted of excess serum submitted for unrelated, clinical purposes to the laboratories. No

blood was collected specifically for this study. Thus, the method of collection does not constitute animal use and did not require review by the University of Illinois Institutional Animal Care and Use Committee. All data were completely anonymized, with identifiers such as names and addresses removed, and unique IDs assigned to the cat populations to maintain confidentiality.

## Study setting

Our study centered on Illinois [Degrees, Minutes, Seconds (DMS) 40˚ 0' 0" N, 89˚ 0' 0" W], a state situated in the Midwest of the United States, with its most populous city, Chicago, positioned on Lake Michigan's southwestern coast (**Fig 1**).

## Serological tests

A blocking enzyme-linked immunosorbent assay (bELISA) for detecting SARS-CoV-2 specific antibody response was developed and validated in-house. The detailed method was described in our previous publication [26]. All the serum samples collected from the cats were initially

**Fig 1. Map highlighting the study area.** Illinois is located in the midwestern United States, which contains 103 counties.

screened by bELISA. The samples with positive bELISA results were subsequently tested by Lumit™ Dx SARS-CoV-2 Immunoassay to confirm the positive status. Lumit™ Dx SARS-CoV-2 Immunoassay is a commercial assay (Promega, Madison, Wisconsin). The lumit assay was conducted following the manufacturer's instructions. For those samples that do not have consistent results from bELISA and Lumit assay, a virus-neutralizing assay was performed in BSL3 laboratory. The detailed method for the virus-neutralizing assay was described previously [10], in which SARS-CoV-2 variant D614G and Omicron were used to test the neutralizing ability of the serum antibody. A total of 244 (out of 1715) samples consistently showed positive results in both bELISA and Lumit assay and were used for subsequent epidemiological analysis (**S1 Table**). Nine samples showed negative results in both Lumit and virus-neutralizing assays and were excluded from the further analysis (**S2 Table**).

## Logistic regression analysis

To assess the impact of the age of cats on the seroprevalence of SARS-CoV-2 antibodies, a logistic regression model was constructed. The predictor variable was represented by the age of cats in months while the outcome variable signified whether the SARS-CoV-2 antibody was detected or not. An odds ratio (OR), 95% confidence intervals, and p-value were calculated for the outcome variable. An OR of < 1 indicated that the probability of SARS-CoV-2 antibody positivity decreased with an increase in age, and if the OR> 1 then the probability of positivity increased with an increase in age. To interpret the results, marginal effects for a series of age intervals were calculated and illustrated in a figure.

## Spatial analysis

A stepwise spatial analysis framework [27] was followed to identify locations and periods with high seroprevalence.

## County-level analysis

All maps for this study were built using ArcGIS Pro version 3.0.3 (Environmental Systems Research Institute, Inc. (ESRI), Redlands, CA, USA).

The analysis was carried out at the county level, a well-defined geographical area used for administrative and statistical purposes in the United States (**Fig 1**). For all spatial analysis, NAD 1983 UTM Zone 16N was used as projection.

For the spatial statistical analysis, the spatial scale was represented by the counties' centroids, and to each centroid, a value representing the seroprevalence in a county (number of positive samples divided by the total samples tested) was linked. Euclidean distance bands were used to measure distances from each county centroid to the neighboring county centroids.

## Disease mapping

A point map was constructed to illustrate the distribution of the county-level seroprevalence of SARS-CoV-2 antibodies in Illinois household cats, using Natural Jenks classification to define the intervals [28].

Spatial interpolation of the seroprevalence of SARS-CoV-2 antibodies in Illinois domestic cats was performed using the Empirical Bayesian Kriging method, which applied a restricted maximum likelihood estimation and constructed several semivariograms to account for the error when estimating the semivariogram [29]. The result of the spatial interpolation was illustrated in an isopleth map.

## Global spatial cluster analysis

The Incremental Spatial Autocorrelation (Global Moran's I) Tool was used to evaluate the global clustering of seroprevalence of SARS-CoV-2 antibodies by assessing a series of incrementally increasing distances and examining the strength of global spatial clustering at each distance [30].

The starting distance was at which each location had at least one neighbor. For each distance, a Moran's I Index value and a z-score and p-value were calculated to test the null hypothesis of spatial randomness. The zone of indifference parameter for the conceptualization of spatial relationships was used for the local and global spatial cluster analysis [30]. The distance band with the highest global clustering (highest Moran's I Index) was selected for the local spatial cluster analysis.

## Local spatial cluster analysis

The Getis-Ord Gi* statistic [31] was used to identify county-level statistically significant hot spots and cold spots. Hot spots signified counties with a high seroprevalence of SARS-CoV-2 antibodies surrounded by counties with high seroprevalence; whereas cold spots indicated counties with low seroprevalence surrounded by counties with low seroprevalence.

## Temporal, spatial, and space-time scan statistic

A retrospective, temporal, spatial, and space-time scan statistic was utilized to identify locations and periods with higher-than-expected seroprevalence of SARS-CoV-2 antibodies using SaTScan software version 9.6 [32].

The smallest spatial scale was represented by the centroid of a county while the time unit was represented by the month and year of SARS-CoV-2 antibody testing. Because the data consisted of two possible outcomes, SARS-CoV-2 antibody positive and negative, a Bernoulli model [33] was used to estimate the relative risk and log-likelihood ratio. A circular scanning window for spatial [34] and a cylinder with a circular spatial base and height relating to time for space-time [35] were used to identify clusters with higher-than-expected SARS-CoV-2 antibodies. The scanning window was set to include 50% of the population and/or 50% of the time at risk. A simulated p-value of ≤0.05 after 999 replications using a Monte Carlo simulation identified significant clusters. Relative risks of counties included within the significant space and space-time clusters were calculated and illustrated in maps to avoid the assumption that the relative risk of seroprevalence is identical throughout a significant cluster.

## City-level analysis

In areas identified in the county-level analysis with high seroprevalence, city-level scan statistic was conducted to identify temporal, spatial, and space-time clusters. The cities' centroids represented the smallest spatial scale, while the month and year of SARS-CoV-2 antibody testing represented the time scale. A Bernoulli model was constructed following the previously described statistical methods.

## Results

### Seroprevalence and distribution of seropositivity of cats across seasons, genders, ages, and breeds

From October 2021 to May 2023, a total of 1,715 household cat serums from 63 counties in Illinois were tested for SARS-CoV-2 specific antibodies. The result showed that 244 (14.23% of 1,715) cats were detected as having specific antibodies against SARS-CoV-2 (**S3 Table**).

**Table 1. Distribution of seropositivity of Illinois household cats across seasons, sex, age, and breed.**

| Factors | | Positive Cases (n) | Total (n) | Positive Rate (%) |
|---|---|---|---|---|
| Season | Winter 2021 (Nov-Jan) | 17 | 248 | 6.85 |
| | Spring 2022 (Feb-Apr) | 58 | 539 | 10.76 |
| | Summer 2022 (May-Jul) | 58 | 263 | 22.05 |
| | Fall 2022 (Aug-Oct) | 61 | 257 | 23.74 |
| | Winter 2022 (Nov-Jan) | 21 | 144 | 14.58 |
| | Spring 2023 (Feb-Apr) | 26 | 233 | 11.16 |
| Sex | Male | 135 | 966 | 13.98 |
| | Female | 106 | 734 | 14.44 |
| Age | Kitten (<1 year) | 8 | 27 | 29.63 |
| | Junior (1–2 years) | 10 | 45 | 22.22 |
| | Adult (3–6 years) | 56 | 309 | 18.12 |
| | Senior (7–10 years) | 59 | 384 | 15.36 |
| | Geriatric (>15 years) | 108 | 939 | 11.5 |
| Breed | Domestic Shorthair | 170 | 1142 | 14.89 |
| | Domestic Longhair | 29 | 208 | 13.94 |
| | Domestic Medium-Hair | 18 | 116 | 15.52 |
| | Maine Coon | 5 | 31 | 16.12 |
| | Siamese | 4 | 30 | 13.33 |

The distribution of seropositivity of cats across season, gender, age, and breed is presented in **Table 1**. Kittens (< 1-year-old) were 29.63% positive, followed by Junior (1–2 years old), 22.22% positive, Adults (3–6 years old) 18.12% positive, Seniors (10–19 years old), 15.36% positive, and Geriatric (>15 years old), 11.5% have the lowest rate of test positivity. Winter, 2021 had the lowest positive rate (n = 17, 6.85% of 248) among all seasons.

We further performed a case-case multivariable logistic regression analysis. The predicted marginal effects of the age and season impact on the probability of SARS-CoV2 antibodies presence in domestic cats are illustrated in **Fig 2**. The results showed that young cats had a higher probability of SARS-CoV-2 infection (OR = 0.996; p = 0.001) as the odds of seropositivity decreased when the age of cats increased. In addition, the third quarter of the year (OR = 3.15; p<0.001) compared to the first quarter had the highest odds of a cat being seropositive. No significant associations were detected with breeds and sexes.

## Disease mapping

We further analyzed the distribution of SARS-CoV-2 seroprevalence in domestic cats across Illinois counties. The county-level seroprevalence ranged from 0 to 100% (**Fig 3A**).

The highest seroprevalence level was in DeKalb, Grundy, Kendall, and White counties, with a 100% positive ratio. The lowest seroprevalence level was observed in 21 counties, including DeWitt, Woodford, Effingham, Wayne, and others, with a 0% positive ratio. In terms of the total number of tests among counties (**S3 Table**), Champaign had the highest number of tests (n = 659), followed by Cook (n = 190), McLean (n = 100), Sangamon (n = 74), and Macon (n = 62).

**Fig 3B** illustrates the spatial interpolation of the seroprevalence of SARS-CoV-2 antibodies in Illinois domestic cats using Empirical Bayesian Kriging analysis. Several regions in Illinois showed an increased seroprevalence, including northern and southern Illinois, with values ranging from 36.98 to 44.02.

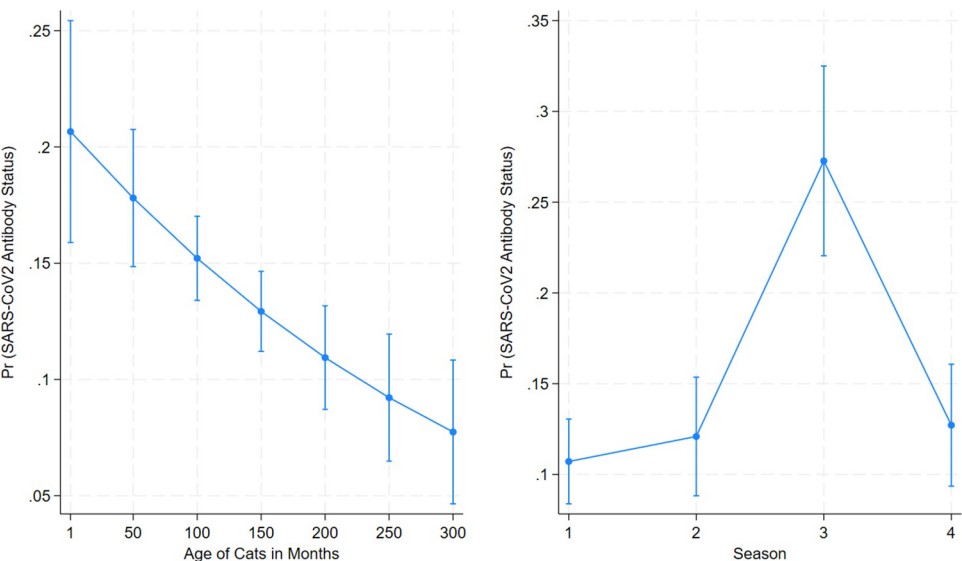

**Fig 2. Effects of age and season on the seroprevalence of SARS-CoV-2 antibody detection in household cats in Illinois, 2021–2023.** Predicted marginal effects calculated from a multivariable logistic regression model. Seropositivity (yes/no) of cat serum samples (n = 1,715) as the outcome variable, while age and season as predictor variables were included in the model.

## Global and local spatial cluster analysis

To determine the extent of clustering of high SARS-CoV2 antibodies in domestic cats across Illinois, global spatial clustering was assessed by using the Incremental Spatial Autocorrelation (Global Moran's I) Tool. As shown in **Fig 4**, one peak (corresponding to the maximum Z-score) was identified at 80.7 km.

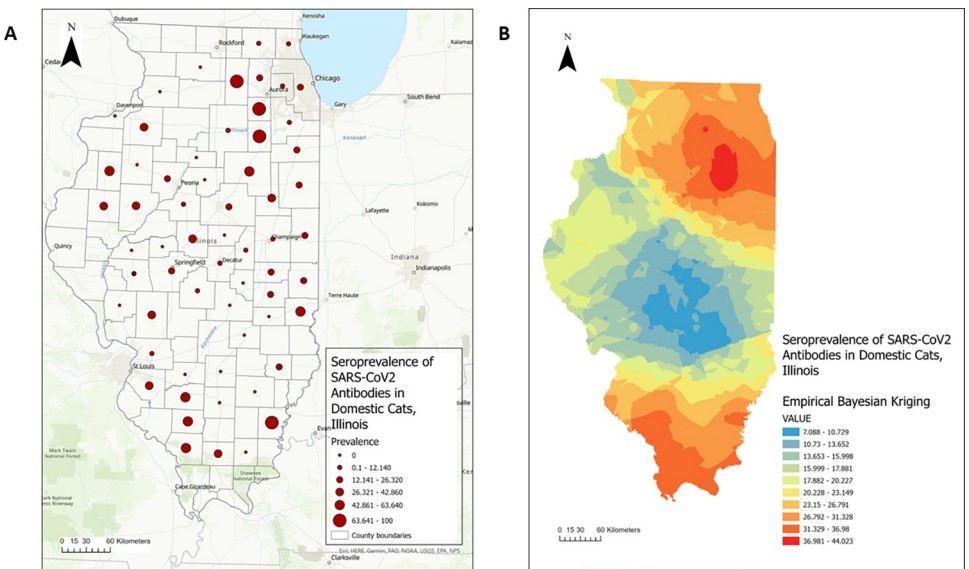

**Fig 3. SARS-CoV-2 antibodies in Illinois household cats during 2021–2023. (A)** Point map illustrating the distribution of seroprevalence by county. **(B)** Isopleth map illustrating the distribution of seroprevalence across Illinois by using the Empirical Bayesian kriging spatial interpolation method.

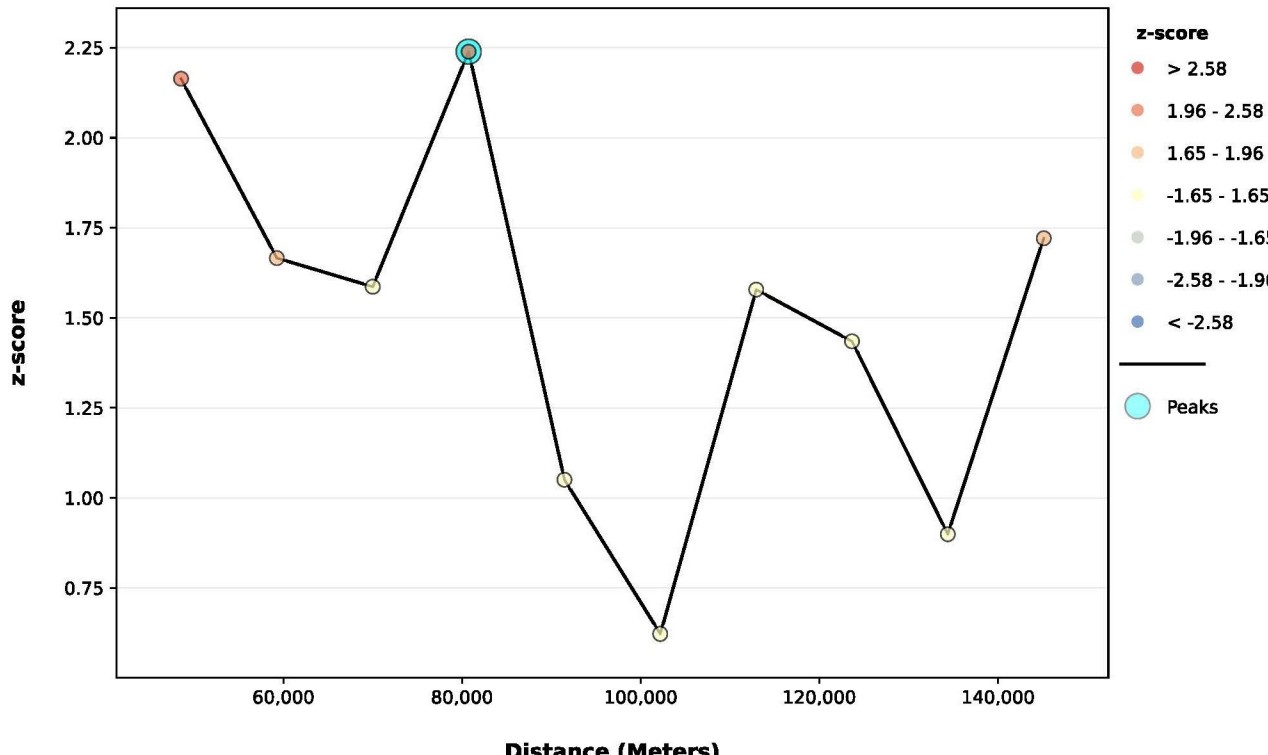

**Fig 4. Incremental spatial autocorrelation analysis for the seroprevalence of SARS-CoV-2 antibodies in Illinois domestic cats.** Results of the Global Moran's I statistic. The default incremental distance was selected as the starting distance that signifies the average distance to each county's nearest neighboring centroid. The color of each point on the graph corresponds to the statistical significance of the z-score values. The peak signal is the distance where the spatial processes influencing clustering are most evident. The zone of indifference conceptualization parameter was used for the analysis. Statistically significant at $p \leq 0.05$.

The high global clustering range suggests a widespread distribution of SARS-CoV2 antibodies in domestic cats across Illinois counties. This distance band and the "zone of indifference" conceptualization parameter were used for the local spatial cluster analysis to determine the global clustering of SARS-CoV2 antibodies (high and low seroprevalence regions).

The Hot Spot (Getis-Ord Gi*) analysis identified 9 counties in northern Illinois with high seroprevalence (hot spots), including LaSalle, Kendall, Grundy, and Kankakee (P-value = 0.01), DuPage and Cook (P-value = 0.05), DeKalb, Kane, and Will (P-value = 0.10) (**Fig 5**).

In addition, in southern Illinois, Williamson County was also identified as a hot spot (P-value = 0.10). A large cold spot (counties with low seroprevalence surrounded by counties with low seroprevalence) was identified in central Illinois that included 7 counties (Logan, De Witt, Macon, Christian, Shelby, Moultrie, and Effingham County, P-value = 0.10).

## Temporal, spatial, and space-time scan statistic analysis

The results of the temporal, spatial, and space-time scan statistics are presented in **Table 2**. A single temporal cluster was identified between June and November 2022, where cats showed higher-than-expected seropositivity for SARS-CoV-2 antibodies. Two spatial clusters (p<0.05) where cats revealed a higher-than-expected -seroprevalence were detected by using the discrete Bernoulli model (**Fig 6A, Table 2**). The primary cluster (SP Cluster 1) was located in northeastern Illinois and included 3 counties (Livingston, Ford, and Grundy) and contained 9 seropositive cases (60% of 15). The second spatial cluster (SP Cluster 2) included 7 northern

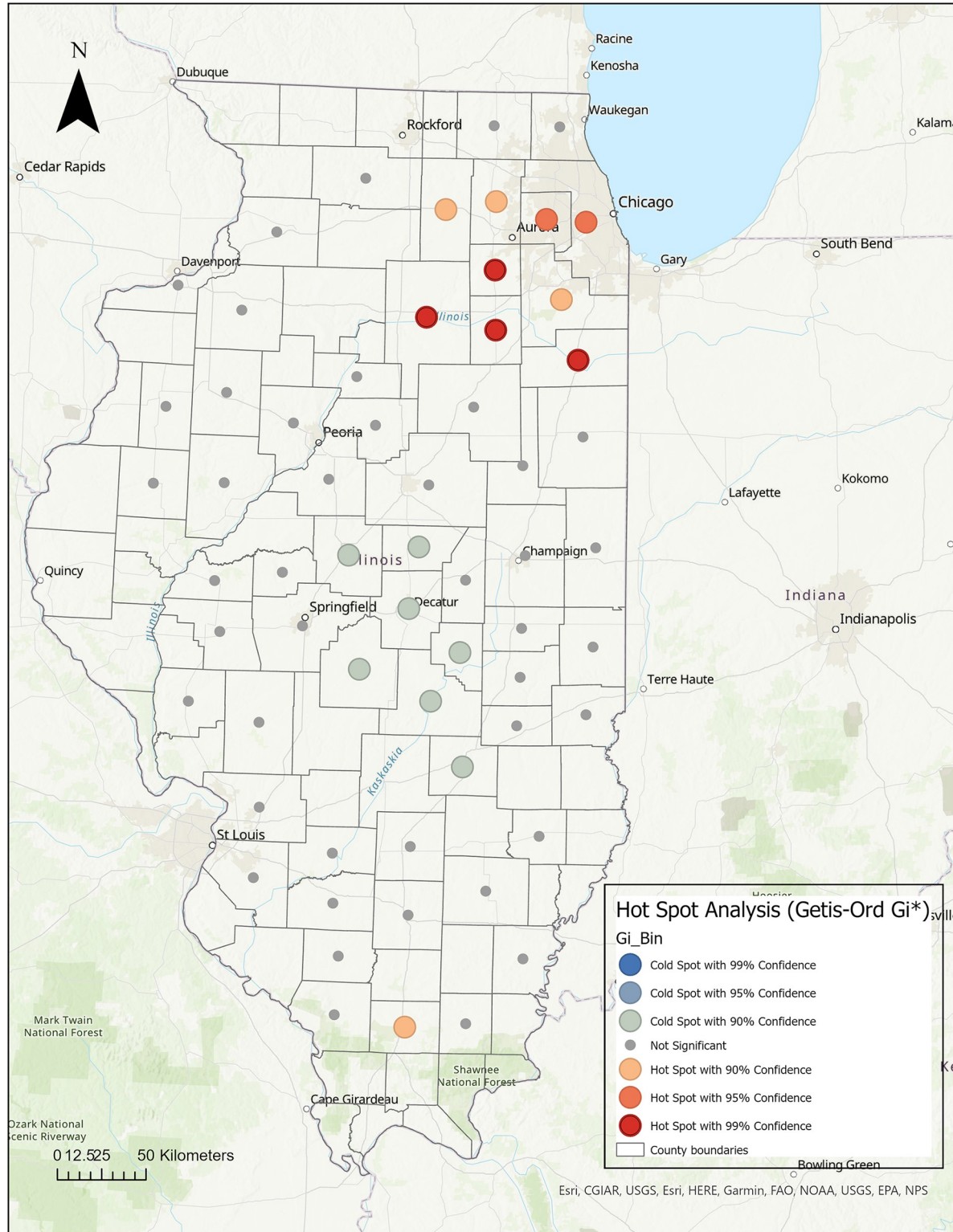

**Fig 5. Hot spot analysis of the seroprevalence of SARS-CoV-2 antibodies in Illinois household cats.** The Getis-Ord Gi* statistic was applied to identify local clusters. Hot spots (red color) signify counties with high seroprevalence surrounded by high seroprevalence counties, while cold spots (blue color) signify counties with low seroprevalence surrounded by counties with low seroprevalence. A Euclidean distance band of 80.7 km, and the zone of indifference conceptualization parameter, were used for the analysis. Statistically significant at p ≤ 0.05.

**Table 2. Spatial and space-time clusters of SARS-CoV-2 infection in household cats in Illinois, United States, 2021–2023.**

| Cluster Type | Cluster | Counties (n) | Radius (km) | Time Frame | Population | Observed Cases | Expected Cases | Relative Risk | Percent cases in the cluster | Log-Likelihood Ratio | P-Value |
|---|---|---|---|---|---|---|---|---|---|---|---|
| **Temporal** | NA | All | NA | 2022/6–2022/11 | 497 | 125 | 70.71 | 2.57 | 25.2 | 31.49 | 0.001 |
| **Spatial** | C1 | 3 | 45.26 | NA | 15 | 9 | 2.13 | 4.34 | 60.0 | 8.49 | 0.006 |
| | C2 | 7 | 89.85 | NA | 28 | 12 | 3.98 | 3.12 | 42.9 | 6.8 | 0.021 |
| **Space-Time** | C1 | 8 | 78.14 | 2022/6–2022/9 | 28 | 18 | 3.98 | 4.80 | 64.3 | 18.87 | <0.001 |
| | C2 | 7 | 64.22 | 2022/7–2022/10 | 196 | 5 | 27.89 | 2.10 | 26. | 11.73 | 0.013 |
| | C3 | 5 | 46.98 | 2022/7–2022/10 | 6 | 6 | 0.85 | 7.18 | 100.0 | 11.76 | 0.013 |

Illinois counties (Jackson, Perry, Williamson, Washington, Jefferson, Saline, and St. Clair), and contained 12 seropositive cases (42.9% of 28). The relative risk within the significant clusters ranged from 0 to 7.05.

The space-time analysis using the Bernoulli model identified three significant (p<0.05) clusters of cats with higher-than-expected SARS-CoV-2 antibodies (**Fig 6B, Table 2**). The primary cluster (ST Cluster 1) occurred between June 2022 and September 2022 in the northeastern part of Illinois (RR = 4.80) and contained 8 counties (Kankakee, Will, Iroquois, Grundy, Livingston, Ford, Kendall, Cook). The second cluster (ST Cluster 2) occurred between July 2022 and October 2022 in the southern part of Illinois (RR = 2.10) and contained 5 counties (Washington, Clinton, Perry, Jefferson, St. Clair). The last cluster (ST Cluster 3) occurred between July 2022 and October 2022 in the central-eastern part of Illinois (RR = 7.18) and contained 5 counties (Edgar, Clark, Douglas, Coles, Vermilion, Cumberland, Champaign). The relative risk within the significant space-time clusters ranged from 0 to 7.13.

### City-level analysis

The city of Chicago and its suburbs were identified in the county-level analysis as an area with high seroprevalence. No purely spatial significant cluster was identified. A temporal cluster was detected between June 2022 and October 2022 where cats showed higher-than-expected seropositivity (Table 3). The space-time analysis identified two significant (p<0.05) clusters of higher-than-expected SARS-CoV-2 antibodies (**Fig 7, Table 3**).

The primary cluster (ST Cluster 1) occurred in June 2022 in the southwestern region of Chicago (RR = 7.75) and contained 20 cities (Braidwood, Wilmington, Channahon, Morris, Minooka, Manhattan, Joliet, Manteno, Bourbonnais, Bradley, Kankakee, Crest Hill, New Lenox, Peotone, Lockport, Frankfort, Plainfield, Romeoville, Momence, and Saint Anne), however, only three cities Morris, Bourbonnais, and Saint Anne had a relative risk larger than one. The second cluster (ST Cluster 2) occurred between July 2022 and September 2022 in the city of Chicago and its suburbs (RR = 6.44) which contained 13 cities (Oak Lawn, Bridgeview, Palos Heights, Blue Island, Orland Park, Cicero, Berwyn, Tinley Park, La Grange Park, Homewood, Darien, Oak Park, and Chicago), however, only five cities had higher than one relative risk, including Tinley Park, Palos Heights, La Grange Park, Berwyn, and Chicago.

### Comparing the seroprevalence of SARS-CoV-2 in cats with the SARS-CoV-2 infections in humans at Illinois

During the 2021–2023 period, the prevalence of SARS-CoV-2 infections in both humans and cats in Illinois, United States, exhibited notable trends. We assessed the potential correlation of

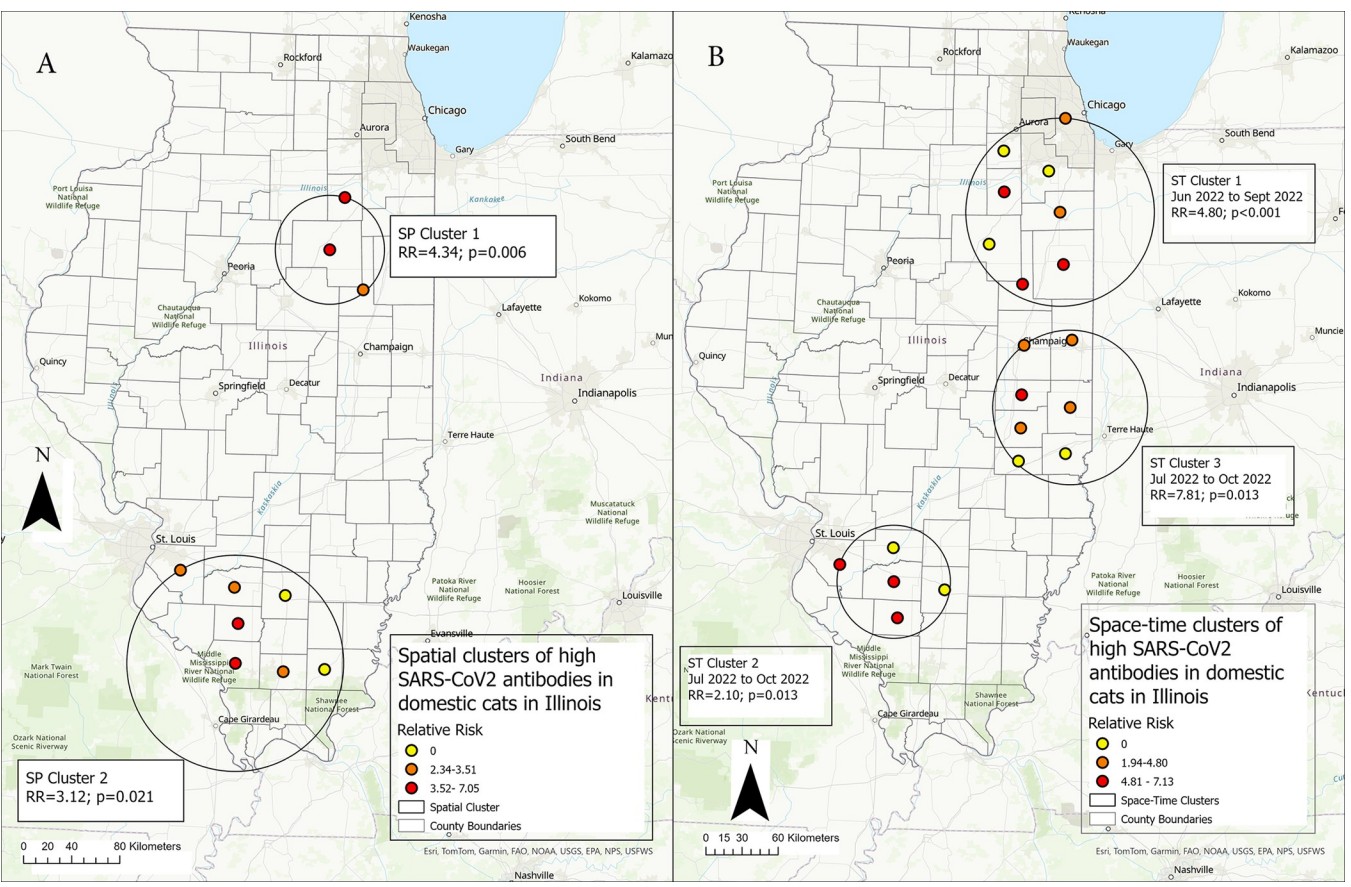

**Fig 6. Spatial and space-time clustering of seroprevalence of SARS-CoV-2 antibodies in Illinois household cats during 2021–2023.** (A) Purely spatial clusters of SARS-CoV-2 antibodies in Illinois household cats. The relative risk (RR) of counties within a spatial cluster where cats had higher than expected SARS-CoV-2 antibodies is shown. The circle represents the location of the cluster. Within the circle, the color of each dot represents the value range of the RR rate in each county. (B) Space-time clusters of SARS-CoV-2 antibodies in Illinois household cats. Retrospective analysis, scanning for clusters with high seroprevalence, using 50% population at risk and 50% period scanning window, 999 Montecarlo permutation, and the Bernoulli model. The circle represents the location of the cluster, and the period of the cluster is also represented. Within the cluster, the color of each dot represents the value range of the RR in each county. Statistically significant at p ≤ 0.05.

SARS-CoV-2 seroprevalence in cats with that of human cases. As shown in the kinetic curve for time course monitoring of antibody response (**Fig 8**), the seroprevalence of SARS-CoV-2 antibody-positive cats reached several peaks, in December 2021, March 2022, April 2022, July 2022, September 2022, October 2022, January 2023, and March 2023. Compared to the reported SARS-CoV-2 positive human cases (nucleic acid-based test), the first SARS-CoV2 antibody prevalence peak in cats overlapped with the human SARS-CoV2 RNA-positive peak from December 2021 to January 2022, while the second SARS-CoV2 antibody prevalence peak in cats appeared about one month after the human positive peak. For the rest of the period, no similarity was found between the positive rates of human and cat cases.

## Discussion

SARS-CoV-2 is known to infect humans and certain animal species. Studies have demonstrated that cats are highly susceptible to SARS-CoV-2 under both experimental and natural infection conditions [36, 37]. Given that domestic cats often live close to their human caretakers and have opportunities to interact with other animals, they may contribute to the ongoing evolution of SARS-CoV-2 [38].

**Table 3. Space-time clusters of SARS-CoV-2 infection in household cats around Chicago city, United States, 2021–2023.**

| Cluster Type | Cluster | Cities (n) | Radius (km) | Time Frame | Population | Observed Cases | Expected Cases | Relative Risk | Percent cases in the cluster | Log-Likelihood Ratio | P-Value |
|---|---|---|---|---|---|---|---|---|---|---|---|
| **Temporal** | NA | All | NA | 2022/6–2022/10 | 43 | 18 | 6.26 | 4.08 | 41.9 | 10.61 | 0.001 |
| **Space-Time** | C1 | 20 | 49.06 | 2022/6–2022/6 | 6 | 6 | 0.87 | 7.75 | 100 | 11.91 | 0.014 |
| | C2 | 13 | 21.94 | 2022/7–2022/9 | 10 | 8 | 1.46 | 6.44 | 80.0 | 11.31 | 0.020 |

We performed serological surveillance of SARS-CoV-2 antibody prevalence in Illinois domestic cats during 2021–2023. Initially, we used the bELISA to screen all the serum samples collected during the study period. For the bELISA positive samples, we used two serological

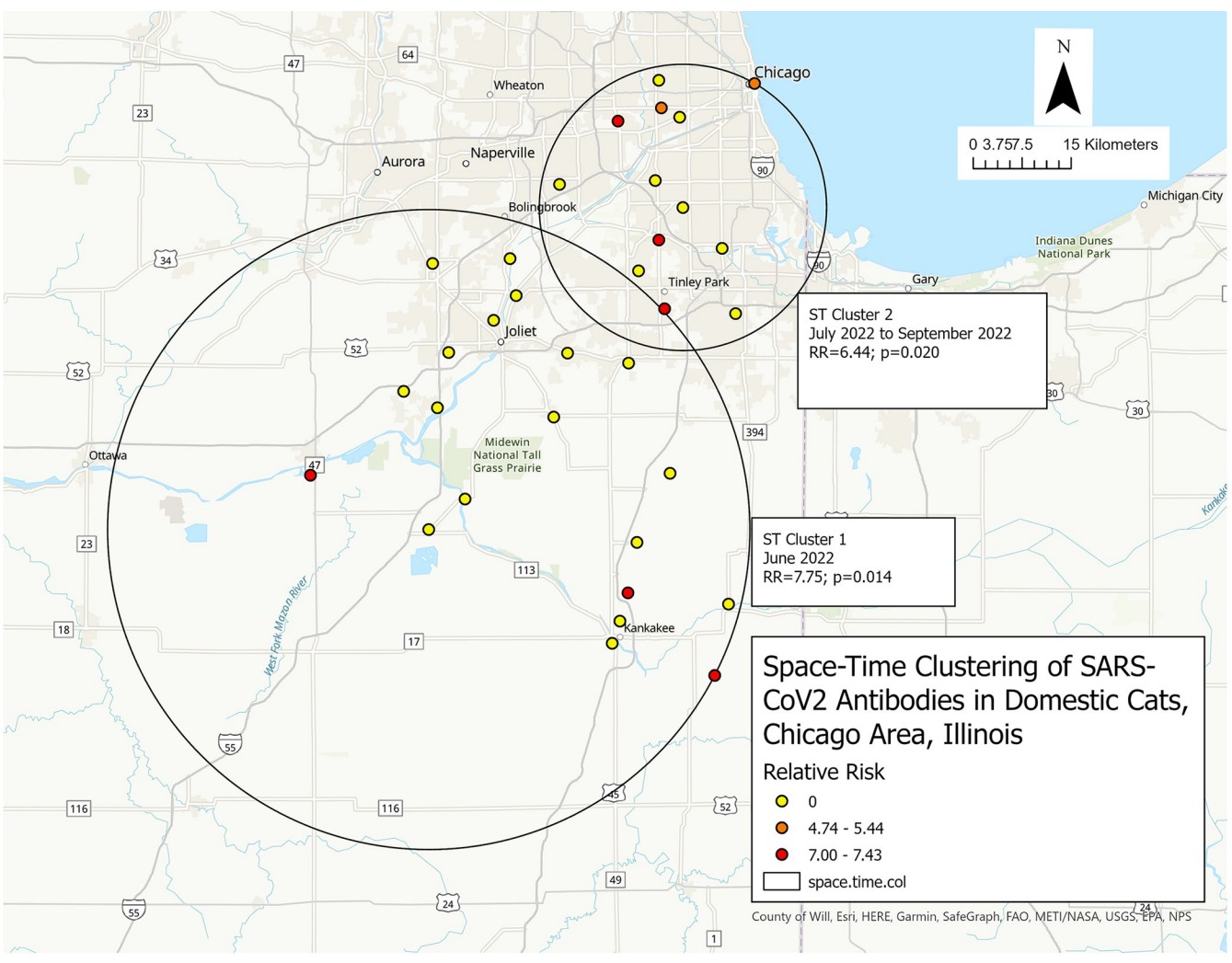

**Fig 7. Space-time clustering of seroprevalence of SARS-CoV-2 antibodies in household cats in the Chicago area during 2021–2023.** Retrospective analysis, scanning for clusters with high seroprevalence, using 50% population at risk and 50% period scanning window, 999 Montecarlo permutation, and the Bernoulli model. The circle represents the location of the cluster, and the period of the cluster is also represented. Within the cluster, the color of each dot represents the value range of the RR in each city. Statistically significant at p ≤ 0.05.

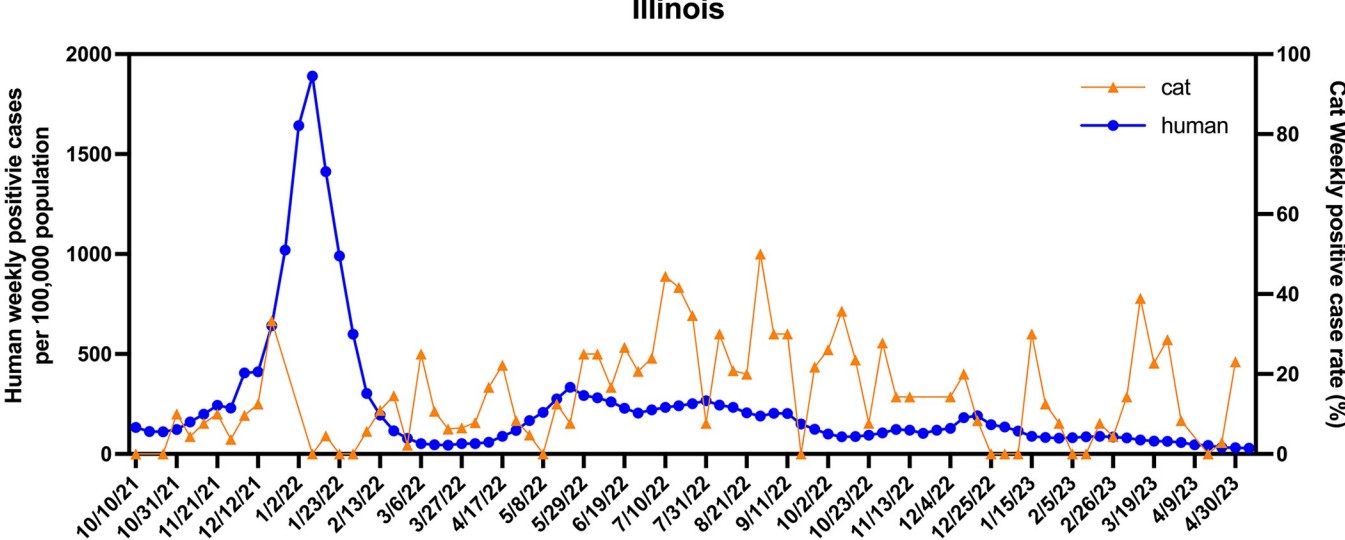

**Fig 8. Prevalence rates of SARS-CoV-2 infections in humans and domestic cats in Illinois during 2021–2022.** The weekly seropositive rate of Illinois domestic cats is shown in the orange line, while the weekly viral nucleic acid test positive rate of reported human cases is shown in the blue line. Note, the peak of human cases during January of 2022, followed by a set of peaks of cat cases starting on February 2022 and continuing for the rest of the study period.

assays (Promega Lumit™ Dx SARS-CoV-2 Immunoassay and virus neutralization assay) to confirm the results. Nine samples showed inconsistent results, which were excluded from the subsequent analysis. The inconsistent results might be due to the different targets between different assays. Our bELISA is targeted for antibodies specific for viral nucleocapsid protein, while Lumit and virus neutralization assays are designed for testing antibodies' response against viral spike protein. On the other hand, the sample quality and test sensitivity could affect the results. Two of the cat serum samples were hemolyzed, which may cause a high background in bELISA test and generate false positive results. Nevertheless, we also performed the analysis with the inclusion of those 9 samples, and the results did not change our conclusions.

To better understand the demographic characteristics of those cats having SARS-CoV-2 specific antibodies, we utilized a case-case multivariable logistic regression analysis to compare the odds of positivity considering seasonality, gender, and age factors. We found that the odds of a cat testing positive for SARS-CoV-2 antibodies decreased as the age increased, with younger cats having the highest while older cats had the lowest seroprevalence, which is opposite to age-related human SARS-CoV2 risks. In human cases, there is an age-related disparity in the prevalence and severity of SARS-CoV-2 infection, in which older people have a higher infection risk due to weaker immune systems. Our study showed the opposite tendency in cats. Young cats have an immature adaptive immune response [39], which may cause them more susceptible to infections. Another possibility is that some of the older cats may have been previously exposed/infected with SARS-CoV-2 or feline coronavirus (FCoV), and they could be protected from the SARS-CoV-2 infection by pre-existing immunity. A recent study demonstrated that both FCoV serotype 1 and 2 infected domestic cats developed cross-reactive antibodies against the SARS-CoV-2 receptor binding domain [40]. Future studies are needed to elucidate the in-depth mechanisms that cause the difference in seroprevalence among various age groups of cats.

To characterize the distribution of SARS-CoV-2 antibodies in Illinois cats, we applied a stepwise analysis that combined both spatial and conventional statistical methods with Geographic Information Systems (GIS). As the first step of spatial analysis, we constructed disease

maps to visualize the geographic distribution of SARS-CoV-2 infections across Illinois's 63 counties. When evaluating the seroprevalence level, we calculated the proportion of positive samples collected from each county across Illinois. One limitation of our approach was that not all Illinois counties were included because from some counties we did not receive samples. To mitigate this issue, we constructed an isopleth map by spatially interpolating the seroprevalence using the empirical Bayesian Kriging method [41]. The results identified areas with high seroprevalence levels of SARS-CoV-2 antibodies in Illinois household cats. However, we found that the distribution of seroprevalence was not even within different counties. In the northern and southern regions of Illinois, the seroprevalence was higher than in other regions. The northern region contains LaSalle, Kendall, Grundy, Kankakee, DuPage, Cook, DeKalb, Kane, and Will counties. This region has the highest human population density in Illinois, which includes the city of Chicago in Cook County, and based on the size of its population, it has the highest estimated number of household cats in Illinois. The southern region in Illinois also displayed a high seroprevalence in both choropleth and isopleth maps. However, this region has a low population density and a low estimated number of cats, and fewer samples were collected from this region (**S3 Table**), which might overestimate the seroprevalence.

The Incremental Spatial Autocorrelation (Global Moran's I) Tool was utilized to examine the global clustering of antibody seroprevalence over ten increasing distance bands. This evaluation of global spatial autocorrelation considered both county locations and their seroprevalence. The concept of "zone of indifference" was used for the analysis, which implies that all counties within a specific distance band receive the maximum weighting, and beyond this distance, the level of influence experiences a rapid decline in weighting as the distance increases [42, 43]. The highest Moran's I Index value, z-score, and p-value were observed at a distance of 80 km, suggesting that SARS-CoV-2 infections were widespread in several areas across Illinois.

We used the Hot Spot Analysis (Getis-Ord Gi*) during the second stage of our spatial analysis. This method identifies areas with high seroprevalence (hot spots; a county with a high seroprevalence surrounded by counties with high seroprevalence) and also detects areas with low seroprevalence (cold spots; a county with low seroprevalence surrounded by counties with low rates) [44]. As presented in the hot/cold spot map, northern Illinois, including LaSalle, Kendall, Grundy, Kankakee, DuPage, Cook, DeKalb, Kane, and Will counties, and southern Illinois containing Williamson County were recognized as statistically significant hot spots, which is consistent with our previous disease mapping results. Similarly, to our previous results, central Illinois was identified as a cold-spot area where cat samples submitted from these counties had a low seroprevalence. As a limitation of our analysis, some regions included counties with low tested sample volumes, and the calculated seroprevalences become unreliable and very high if several positive cases were detected in these regions [27].

At the third stage of our spatial analysis, we utilized temporal, spatial, and space-time scan statistics to add extra information on the location and time frame of the distribution of seroprevalence of SARS-CoV-2 positive antibodies in Illinois household cats. First, we employed a purely spatial scan statistic, which presumes that the seroprevalence follows a Bernoulli distribution (i.e., SARS-CoV-2 antibody detected in a cat in a county versus not detected). Two high seroprevalence spatial clusters were identified in the northeast region of Illinois, where 60% of the samples tested positive, and the southern region of Illinois, where 42.9% of the samples tested positive. These clusters overlapped with the area identified by the Hot Spot method. We also detected three high seroprevalence space-time clusters in the northeast, central-east, and southern regions of Illinois. Space-time cluster 1 (ST 1), occurring between June 2022 and September 2022, and space-time cluster 2 (ST 2), occurring between June 2022 and September 2022 overlapped with the clusters detected by the spatial scan and Hot Spot analysis, highlighting the importance of these areas. The third space-time cluster (ST 3), which occurred between

July 2022 and October 2022) was detected in the central-eastern Illinois region. These space-time clusters beside the locations provided a time component, suggesting that there were peaks in seroprevalence that occurred during these periods.

Chicago is Illinois' most populous city. Since it is a global transportation hub with dense urban environment, it is crucial to understand the SARS-CoV-2 epidemiology in this area. Through temporal and space-time analysis, we identified a temporal cluster that occurred between June 2022 and October 2022. This time frame was consistent with the temporal cluster identified at the county-level analysis. We also find 2 space-times clusters with higher-than-expected seroprevalence. Space-time cluster 1 (ST 1), which occurred in June 2022, included 20 cities and 3 of them have local relative risk between 7 and 7.43. The space-time cluster 2 (ST 2) which occurred between July 2022 and September 2022 overlapped with the cluster identified at the county-level analysis and contained 5 cities with higher than 5 relative risks. These patterns offer crucial insights for targeted public health interventions and future surveillance efforts in densely populated areas.

Interestingly, there was a reported SARS-CoV-2 human infection peak that occurred in Illinois from December 2021 to January 2022. In our study period, we detected the first cat seropositive peak in late December 2021, while the second cat seropositive peak appeared about one month after the human positive peak. We suspected that there might be some possible transmissions between humans and cats during that period. Especially, the second seropositive peak could be related to human SARS-CoV-2 infections because about 14 days after being infected with SARS-CoV-2, the antibody level is detectable by our serological testing method [10, 26], suggesting a possible human-to-cat transmission. However, we did not find any similarity between the positive case rates of human infections and cat seropositivity for the rest of the study period (between March 2022 and May 2023). Future studies are warranted to follow up on this finding to assess the potential transmission risk between humans and cats.

Before interpreting our study results, a few limitations should be noted. The blood samples were obtained from samples submitted to the Clinical Pathology Laboratory at the University of Illinois Veterinary Diagnostic Laboratory (Urbana, IL) and its satellite lab at the Veterinary Specialty Center (Bannockburn, IL), located in a suburb of the city of Chicago, and the number of samples received from veterinary clinics closest to these locations might be higher. However, Chicago is the largest city in Illinois and has the highest estimated number of cats which might mitigate this effect. In addition, for the serological data, the time of seropositivity is dependent on the time of sample collection and without knowing the duration of seropositivity, the interpretation of temporal trends should be made with caution because the time of exposure and the time of seropositivity does not correlate.

In conclusion, our analysis of SARS-CoV2 antibody seroprevalence in Illinois domestic cats during 2021–2023 identified northeast and southwest Illinois regions with increased seroprevalence of SARS-CoV-2 antibodies among domestic cats, and this increase occurred between June and October 2022. The susceptibility of the cats to SARS-CoV-2 infection appeared to be related to the age and the time of the year. This information helps aid public health stakeholders in developing effective prevention and control measures.

## Supporting information

**S1 Table. ELISA and Lumit™ Dx SARS-CoV-2 Immunoassay results for cat serum samples.** (DOCX)

**S2 Table. ELISA, virus-neutralizing assay, and Lumit™ Dx SARS-CoV-2 Immunoassay results for cat serum samples with inconsistent results.** (DOCX)

**S3 Table. The number of seropositive and total tested samples from household cats in each county of Illinois, 2021–2023.**
(DOCX)

# Acknowledgments

We thank Dr. Tony Vanden Bush from Promega for providing the Lumit™ Dx SARS-CoV-2 Immunoassay kits.

**NOTE:** A preprint of this article has been previously published [Chen et al., 2023, medRxiv, 2023. doi: https://doi.org/10.1101/2023.06.21.23291564.].

# Author Contributions

**Conceptualization:** Chi Chen, Csaba Varga, Ying Fang.

**Data curation:** Chi Chen.

**Formal analysis:** Chi Chen, Csaba Varga.

**Funding acquisition:** Mathias Martins, Diego G. Diel, Ying Fang.

**Investigation:** Chi Chen, Mathias Martins, Mohammed Nooruzzaman, Dipankar Yettapu, Diego G. Diel, Jennifer M. Reinhart, Ashlee Urbasic, Hannah Robinson, Csaba Varga, Ying Fang.

**Methodology:** Mohammed Nooruzzaman, Csaba Varga, Ying Fang.

**Project administration:** Diego G. Diel, Ying Fang.

**Resources:** Mathias Martins, Diego G. Diel, Jennifer M. Reinhart, Ashlee Urbasic, Hannah Robinson, Ying Fang.

**Supervision:** Ying Fang.

**Visualization:** Csaba Varga.

**Writing – original draft:** Chi Chen.

**Writing – review & editing:** Mathias Martins, Mohammed Nooruzzaman, Dipankar Yettapu, Diego G. Diel, Jennifer M. Reinhart, Ashlee Urbasic, Hannah Robinson, Csaba Varga, Ying Fang.

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
