## [Decision Letter · Decision Letter 0]

18 Dec 2023

PONE-D-23-31869Spatial and temporal clustering of anti-SARS-CoV-2 antibodies in Illinois household cats, 2021- 2023PLOS ONE

Dear Dr. Varga,

Thank you for submitting your manuscript to PLOS ONE. After careful consideration, we feel that it has merit but does not fully meet PLOS ONE’s publication criteria as it currently stands. Therefore, we invite you to submit a revised version of the manuscript that addresses the points raised during the review process. Please answer carefully to all the comments from the reviewer.

We look forward to receiving your revised manuscript.

Kind regards,

Pierre Roques, Ph.D.

Academic Editor

PLOS ONE

“We thank Dr. Tony Vanden Bush from Promega for providing the Lumit™ Dx SARS-CoV-2 Immunoassay kits. This project was supported by the National Institute of Health [Grant #R01Al166791, to Y. Fang (PI) and D. Diel (MPI)].”

4. We note that Figures 1, 3, 5, 6 and 7 in your submission contain [map/satellite] images which may be copyrighted. All PLOS content is published under the Creative Commons Attribution License (CC BY 4.0), which means that the manuscript, images, and Supporting Information files will be freely available online, and any third party is permitted to access, download, copy, distribute, and use these materials in any way, even commercially, with proper attribution. For these reasons, we cannot publish previously copyrighted maps or satellite images created using proprietary data, such as Google software (Google Maps, Street View, and Earth). For more information, see our copyright guidelines: http://journals.plos.org/plosone/s/licenses-and-copyright.

1. You may seek permission from the original copyright holder of Figures 1, 3, 5, 6 and 7 to publish the content specifically under the CC BY 4.0 license. 

Additional Editor Comments:

Sorry for the delay, but it seems that the survey of anti-sars-CoV-2 in cats is not in the current top priority of veterinarian, even those who publisched on the subject previously.

Reviewers' comments:

Reviewer's Responses to Questions

**Comments to the Author**

1. Is the manuscript technically sound, and do the data support the conclusions?

Reviewer #1: Yes

2. Has the statistical analysis been performed appropriately and rigorously? 

Reviewer #1: Yes

3. Have the authors made all data underlying the findings in their manuscript fully available?

Reviewer #1: Yes

4. Is the manuscript presented in an intelligible fashion and written in standard English?

Reviewer #1: Yes

5. Review Comments to the Author

Reviewer #1: In the manuscript, the authors described the spatio-temporal distribution of SARS-CoV-2 serological results in cats in Illinois. The study is interesting, as it combines geographical, spatial and biological analyses. Through these multidisciplinary approaches, the authors pointed some spatial and geographical occurences. It also clearly support the hypothesis of a SARS-CoV-2 transmission from human toward cat.

Introduction section

Line 69: Dogs can be also infected by SARS-CoV-2. Please add this animal species in your list and justify why in this study, the authors choose to focus only on cat infections.

Result section

The cat sera were tested with different methodologies for the presence of SARS-CoV-2 antibody: ELISA, Luminex and seroneutralisation (D614G and Omicron). Please indicate the results for each methodologies in a supplemental table. Moreover, it would be intersting to add the titer of seroneutralisation for each virus tests.With these results, the authors could evaluate by which variant the cats have been infected. It would be an interesting supplementary information.

Line 368: Specify whether you reported the human cases in Illinois or in United States in figure 8

Discussion section

Lines 400-402: In this study, the authors found the higesht seropositivity in young cats. In the discussion, they should add the possibility that older cats may have been previously infected by SARS-CoV-2. They could also discuss the possiblity of cross-protection or not by previous feline coronavirus infection

6. PLOS authors have the option to publish the peer review history of their article (what does this mean?). If published, this will include your full peer review and any attached files.

Reviewer #1: No

---

## [Author Response · Author response to Decision Letter 0]

18 Jan 2024

Responses to Journal requirements:

Response: The manuscript was re-formatted to meet PLOS ONE's style requirements

2. Response: The Funding Statement should be updated to: “This project was supported by the National Institute of Health [Grant #R01Al166791, to Y. Fang (PI) and D. Diel (MPI)].”

3. Please include your full ethics statement in the ‘Methods’ section.

Response: We included the following: ” Samples consisted of excess serum submitted for unrelated, clinical purposes to the laboratories. No blood was collected specifically for this study. Thus, the method of collection does not constitute animal use and did not require review by the University of Illinois Institutional Animal Care and Use Committee” (Lines 106-110).

4. We note that Figures 1, 3, 5, 6, and 7 in your submission contain [map/satellite] images which may be copyrighted.

Response: We used base maps provided by Arc GIS Pro and/or ArcGIS Online maps hosted by Esri. The appropriate attribution was provided at the bottom of each map.

Response to Reviewer 1’s comments.

Reviewer #1: In the manuscript, the authors described the spatio-temporal distribution of SARS-CoV-2 serological results in cats in Illinois. The study is interesting, as it combines geographical, spatial and biological analyses. Through these multidisciplinary approaches, the authors pointed some spatial and geographical occurences. It also clearly support the hypothesis of a SARS-CoV-2 transmission from human toward cat.

Introduction section

Line 69: Dogs can be also infected by SARS-CoV-2. Please add this animal species in your list and justify why in this study, the authors choose to focus only on cat infections.

Response: “Dogs” was added to the list of animals that can be infected with SARS-CoV2 (lines 67 and 71). The following sentences and references were added to justify why this study focuses only on cat infections (lines 70-76).

“SARS-CoV-2 has been reported to be transmitted from humans to animals, including domestic cats and dogs (15-18). Previous studies showed that SARS-CoV-2 did not replicate well in dogs, but was able to efficiently replicate in domestic cats and further transmit the virus to naive contact animals (19). A recent report provided evidence that domestic cats could transmit SARS-CoV-2 to humans (15). Since domestic household cats have close contact with humans, it is important to understand the prevalence of SARS-CoV-2 exposure in household cat populations.”

Result section

The cat sera were tested with different methodologies for the presence of SARS-CoV-2 antibody: ELISA, Luminex and seroneutralisation (D614G and Omicron). Please indicate the results for each methodologies in a supplemental table. Moreover, it would be intersting to add the titer of seroneutralisation for each virus tests. With these results, the authors could evaluate by which variant the cats have been infected. It would be an interesting supplementary information.

Response: In our study, we initially screened all the cat samples by bELISA, and the bELISA positive samples were subsequently confirmed by Lumit assay. Due to the restrictions working in BSL3 lab, we only performed virus-neutralizing tests on those samples having inconsistent results between bELISA and Lumit assay. 

To address this comment, we added Table S1 as a supplementary table to show the results of different antibody tests. We also clarified our antibody testing procedure in the Materials and Methods section (lines 118-133).

Line 368: Specify whether you reported the human cases in Illinois or in United States in figure 8

Response: We clarified that we referred to the human cases in Illinois. 

Discussion section

Lines 400-402: In this study, the authors found the higesht seropositivity in young cats. In the discussion, they should add the possibility that older cats may have been previously infected by SARS-CoV-2. They could also discuss the possiblity of cross-protection or not by previous feline coronavirus infection

Response: We appreciate your excellent suggestion. The following sentences and reference were added in the discussion section (line 405-411):

“Another possibility is that some of the older cats may have been previously exposed/infected with SARS-CoV-2 or feline coronavirus (FCoV), and they could be protected from the SARS-CoV-2 infection by pre-existing immunity. A recent study demonstrated that both FCoV serotype 1 and 2 infected domestic cats developed cross-reactive antibodies against the SARS-CoV-2 receptor binding domain (40). Future studies are needed to elucidate the in-depth mechanisms that cause the difference in seroprevalence among various age groups of cats."

---

## [Editor Report · Decision Letter 1]

9 Feb 2024

Spatial and temporal clustering of anti-SARS-CoV-2 antibodies in Illinois household cats, 2021- 2023

PONE-D-23-31869R1

Dear Dr. Varga,

We’re pleased to inform you that your manuscript has been judged scientifically suitable for publication and will be formally accepted for publication once it meets all outstanding technical requirements.

Kind regards,

Pierre Roques, Ph.D.

Academic Editor

PLOS ONE
---

## [Editor Report · Acceptance letter]

8 Apr 2024

PONE-D-23-31869R1 

PLOS ONE

Dear Dr. Varga, 

I'm pleased to inform you that your manuscript has been deemed suitable for publication in PLOS ONE. Congratulations! Your manuscript is now being handed over to our production team.

Kind regards, 

on behalf of

Dr. Pierre Roques 

Academic Editor

PLOS ONE